# MaskedKD: Efficient Distillation of Vision Transformers with Masked Images

## Abstract

Knowledge distillation is an effective method for training lightweight models, but the cost of acquiring teacher supervisions on training samples is often significant. Such supervision cost can be overwhelmingly large when we distill from large-scale proprietary models, such as vision transformers (ViTs). We present MaskedKD, a simple yet effective strategy that can significantly reduce the teacher supervision cost, without sacrificing the student accuracy or requiring direct access to (potentially proprietary) teacher. Specifically, MaskedKD diminishes the cost of running teacher at inference by masking a fraction of image patch tokens fed to the teacher, and therefore skipping the computations required to process those patches. The mask locations are selected to prevent masking away the core features of an image that the student uses for prediction. This masking mechanism operates based on some attention score of the student, which is already computed during the student forward pass, and thus incurs almost no additional computation. Our experiments show that MaskedKD dramatically reduces the teacher supervision cost, saving up to $50\%$ teacher FLOPs without student accuracy drop.

## 1 Introduction

We develop a simple technique that can save tremendous amount of computations required to distill knowledge from large vision transformers (ViTs) (Dosovitskiy et al., 2020). Large-scale ViTs are becoming increasingly popular as a vision backbone, but it is difficult for an individual user to deploy such models to their own device. Modern ViTs are typically *too large* to fit into on-device memories, e.g., ViT-22B (Dehghani et al., 2023), and are often *proprietary*, provided only as service APIs with no public access to model parameters (OpenAI, 2023). Under this context, the classic idea of *knowledge distillation* has re-gained attention, as it allows the user to train well-performing lightweight model by utilizing the prediction of large-scale models without requiring parameter-level access (Hinton et al., 2015). Several works have shown that distilling from large teacher ViTs significantly improves the performance of small/compressed students (Wang et al., 2022; Wu et al., 2022a; Hao et al., 2022; Yu et al., 2022; Li et al., 2022b), often by using only teacher predictions without any additional use of intermediate features (thus working with proprietary teachers) (Wu et al., 2022b).

However, the computational cost of acquiring teacher supervision (i.e., *supervision cost*) can be prohibitively expensive. One needs to make multiple predictions for training samples on the teacher ViT, which typically requires even greater computation than the computations required to process the student (see left of Fig. 1). For example, the total supervision cost distilling from ViT-G scale teachers on ImageNet is approximately $10^4$ TPUv3-days (Zhai et al., 2022); this may translate into prohibitively expensive service fee for proprietary teachers. Existing works attempt to relieve the situation by pre-computing teacher's predictions for all training data and re-using them at the distillation (Yun et al., 2021; Shen and Xing, 2022). However, this approach not only require ample storage to save teacher predictions, but also tend to degrade student accuracy, due to their somewhat limited ability to account for various data augmentations, e.g., *mixup* or RandAugment; distilling with teacher's predictions on unaugmented sample to the student seeing augmented data is detrimental to performance (Beyer et al., 2022; Shen and Xing, 2022).

In this paper, we propose a very simple yet effective approach called MaskedKD (Masked Knowledge Distillation) to cut down the ViT supervision cost quite dramatically without degrading the student accuracy. Precisely, MaskedKD masks out a fraction of image patch tokens given as the teacher input

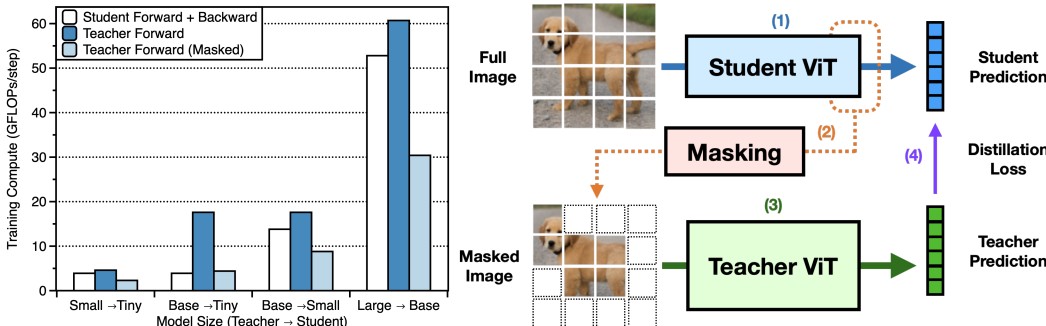

Figure 1: (⇐) **Supervision cost vs. student training cost.** We compare per-step supervision cost of the teacher ViT that sees full or masked images, with the training FLOPs of the student ViT; we mask the teacher input to a point where there is no student accuracy drop. Supervision cost is larger than student FLOPs, and masking can save a great amount. (⇒) **MaskedKD illustrated.** MaskedKD works in four steps: **(1)** Student predicts on the full image. **(2)** Mask the image with the SIPS score. **(3)** Teacher predicts for the masked image. **(4)** Match the teacher and student logits.

and thus skips the computations therein (see right of Fig. 1). This reduces the teacher forward cost without requiring any architectural change or retraining of the teacher ViT, and thus can be used to distill from proprietary teacher models whose model we cannot access.

Quite surprisingly, we find that, given the right masking strategy, masking out a substantial fraction of teacher input patches (25–50%) does not degrade the *supervision quality* of the teacher: The teacher accuracy drops, but the student accuracy remains the same, or even sometimes slightly higher. The masking strategy of MaskedKD operates on the following principles, which we find to be essential for achieving the best compute-efficiency without any student performance drop.

- **Mask teacher, not student:** For supervised knowledge distillation (which is our focus), masking only the teacher model is critical for performance; masking the student degrades the performance even at a very low masking ratio. This strategy critically differs from standard applications of masking in self-supervised learning (or self-distillation) literature, where the student always sees masked input to imitate the outputs of weight-tied teacher that sees full or masked inputs (Caron et al., 2021; Assran et al., 2022; Chen et al., 2022; Li et al., 2022a; Zhang et al., 2023).

- **Student guides teacher what to teach:** To mask the teacher input, we propose a student-informed patch saliency (SIPS) metric, which is designed to prevent masking away the core features that the student uses for prediction. The SIPS metric re-uses the attention scores obtained during the student forward process, and thus adds almost no extra computation. Empirically, we observe that SIPS is well-correlated with the performance of the distilled student and significantly outperforms random masking criterion (used in, e.g., He et al. (2022)), suggesting that the student's guidance is indeed useful in improving the supervision quality of the teacher.

- **Mask at input, not in the middle:** We propose to reduce the number of tokens at the model input (by masking the teacher input), instead of gradually removing tokens in the intermediate layers, as is common in token pruning/merging literature (Goyal et al., 2020; Bolya et al., 2023). This is for two reasons: First, this makes MaskedKD applicable for distilling from proprietary teachers with no public access to model itself. Second, input-masked teachers have a better supervision quality (student accuracy) than the teachers accelerated by standard gradual token removal to have similar inference FLOPs (Bolya et al., 2023). Our analysis shows that input-masked teachers not only teach better, but also tend to retain better prediction quality in the low-PSNR regime.

Throughout our experiments, we find that MaskedKD can decrease the number of input patches given to the teachers by 25–50% without degrading the student accuracy, under diverse choice of teacher/students and base ViT distillation algorithms. This leads to a corresponding decrease in the supervision cost, i.e., the computation required to acquire supervisions from the teacher. We also find that MaskedKD can be applied to transformers for audio data (Gong et al., 2021) and self-supervised learning scenarios, such as DINO (Caron et al., 2021).

## 2 RELATED WORK

**Distilling ViTs.** Most existing literature on ViT distillation focuses on what information should we transfer from the teacher to help training the student ViT. An early work by Touvron et al. (2021a) considers distilling the architectural bias of convolutional networks to enhance the student's data-efficiency. More recent works consider distilling patch-level information of large-scale ViTs to enhance the student performance, e.g., the class-patch attention score (Wang et al., 2022) attention score distribution (Wu et al., 2022a), weight information of self-attention modules (Zhang et al., 2022), or the configuration of the patch manifold (Hao et al., 2022). The present paper also considers a distillation of a ViT to another ViT. Unlike prior work, however, we focus on reducing the supervision cost of the teacher model by exploiting the ViT structure, by having the student guide the teacher.

**Masking as a self-supervision.** Masking has been actively studied in the self-supervised learning (SSL) literature as a pretext task: the model takes masked image as an input and trains to predict missing pixels (He et al., 2022) or to make similar predictions with a weight-tied model that sees the full image (Assran et al., 2022); the latter can be viewed as a form of *self-distillation* (Grill et al., 2020), by treating the weight-tied model as a teacher and the original model as a student. Following the works, many work also proposed various self-distillation algorithms that use masking for SSL (see Table 1 for partial summary and Peng et al. (2023) for an overview). In such works, however, the focus is to maximize the SSL performance rather than to reduce the computation. To our knowledge, none of the works exclusively studies the computational benefit of masking for distillation, *decoupled* from how well the masking serves as the pretext task. In this work, we find that many common masking practices in SSL is suboptimal for reducing supervision cost in supervised distillation.

**Token removal.** Token removal for reducing the computational burden of transformers has been pioneered by (Goyal et al., 2020) for language models, and much efforts have followed to design similar methods for ViTs. DynamicViT (Rao et al., 2021), A-ViT (Yin et al., 2022), and AdaViT (Meng et al., 2022) give algorithms to train a model that gradually removes intermediate tokens as the layer goes deeper, dramatically reducing the inference cost of the model. Instead of completely discarding tokens, Kong et al. (2021), Liang et al. (2022), and Marin et al. (2023) propose to *combine* intermediate tokens into another, instead of removing them. Bolya et al. (2023) introduce a drop-in token merging module that can be used to enhance the inference efficiency without any additional training steps. These works focus on preserving the predictive quality of a model. Our paper, in contrast, focuses on preserving the *supervision quality* of a teacher model.

**Saving the supervision cost.** A recent line of work attempts to reduce the supervision cost by re-using teacher supervisions. In particular, Shen and Xing (2022) draw inspirations from ImageNet re-labeling technique (Yun et al., 2021) to pre-compute teacher model's predictions on multiple random crops of training samples; the crop information and the pseudo-labels (teacher predictions) are stored as additional attributes of the sample. Then, the student is trained by drawing randomly cropped data and corresponding pseudo-labels, and using distillation loss on these samples to update the model. However, this approach has a limited applicability for the cases where more diverse data augmentations are used, leading to a degraded student accuracy (Beyer et al., 2022; Shen and Xing, 2022). On the other hand, the proposed MaskedKD saves the supervision cost by reducing the teacher inference cost directly, applicable to the standard on-the-fly distillation scenario.

Table 1: **Comparison with masking-based SSL algorithms.** We compare the masking strategy of MaskedKD with SSL algorithms that use a form of mask-based distillation as a pretext task. MaskedKD differs from the works in many technical aspects. (★: student sees many random crops.)

| | Distillation setup | | Masked | | Masking Criterion | Teacher type |
|---|---|---|---|---|---|---|
| | Super. dist. | Self-super. dist. | Teacher | Student | | |
| MAE (He et al., 2022) | ✗ | ✗ | - | ✔ | Random | Pixel |
| DINO (Caron et al., 2021) | ✗ | ✔ | ✗ | ★ | Random | EMA |
| MSN (Assran et al., 2022) | ✗ | ✔ | ✗ | ✔ | Random | EMA |
| MaskFeat (Wei et al., 2022) | ✗ | ✔ | ✗ | ✔ | Random | HOG/DINO |
| SdAE (Chen et al., 2022) | ✗ | ✔ | ✔ | ✔ | Random | EMA |
| PCAE (Li et al., 2022a) | ✗ | ✔ | ✔ | ✔ | Activation | EMA |
| ccMIM (Zhang et al., 2023) | ✗ | ✔ | ✔ | ✔ | Shared Attn. | EMA |
| MaskedKD (Ours) | ✔ | ✗ | ✔ | ✗ | SIPS | Supervised |

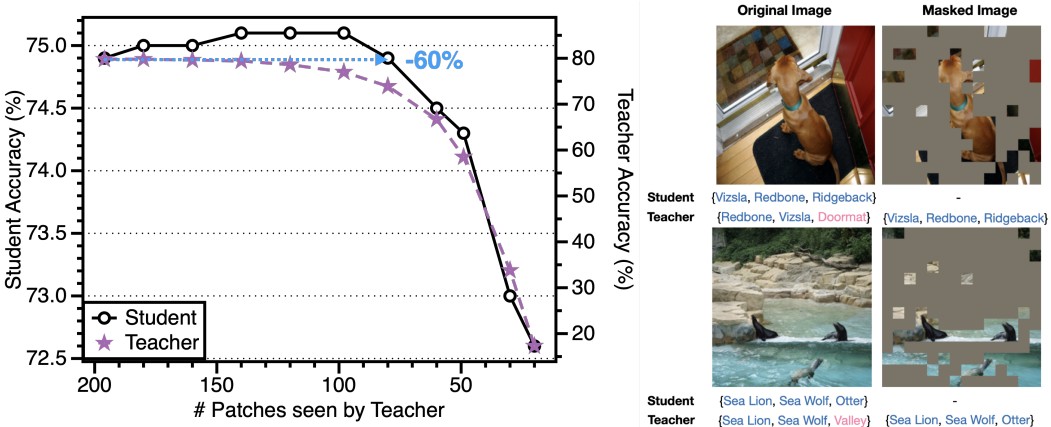

Figure 2: (⇐) **Accuracy vs. # patches seen.** We distill DeiT-Small teacher into DeiT-Tiny student, using MaskedKD with various masking ratio. By masking the teacher, the teacher accuracy monotonically decreases. In contrast, the student accuracy slightly increases first, and then decreases after masking over 50% of the patches. (⇒) **Top-3 predictions of the teacher and the student.** Predictions in blue correspond to the foreground objects in the image, and predictions in red correspond to the context attributes. The masked teacher tends to make more similar predictions to the student.

# 3 METHOD: MASKED KNOWLEDGE DISTILLATION

We present masked knowledge distillation (MaskedKD), a simple yet effective approach for reducing the ViT supervision cost. In a nutshell, MaskedKD masks off a fixed fraction of input image patches, and feeds the masked image through the teacher; the masks are located in a way that the teacher can provide supervisions on the patch which the student utilizes the most to make its own predictions; we provide more details about the masking mechanism in Section 3.1.

More concretely, let $f_S$ and $f_T$ be the student and teacher ViTs, respectively. Given an image-label pair $(x, y)$, MaskedKD generates a masked version of the input image $x_{\mathtt{mask}}$ and train the student model with a combination of the cross-entropy and distillation loss. In other words, we minimize

$$\mathcal{L} = \mathcal{L}_{\mathrm{CE}}(f_S(x), y) + \lambda \cdot \mathcal{L}_{\mathrm{KD}}(f_S(x), f_T(x_{\mathtt{mask}})), \tag{1}$$

where $\mathcal{L}_{\mathrm{CE}}$ denotes the cross-entropy loss, $\mathcal{L}_{\mathrm{KD}}$ denotes the distillation loss, and $\lambda \geq 0$ is a hyperparameter for balancing two losses. The distillation loss $\mathcal{L}_{\mathrm{KD}}$ can be chosen flexibly depending on the base distillation algorithm we want to use, e.g., the KL-divergence for the classic logit distillation (Hinton et al., 2015) or the $\ell_2$ distance between activations (Wu et al., 2022a).

The key difference between standard knowledge distillation and the MaskedKD (Eq. (1)) is that the latter uses the masked teacher input $f_T(x_{\mathtt{mask}})$ instead of $f_T(x)$. This change has following effects:

- **Lowering the supervision cost.** By masking the image, the computations associated with the masked tokens can be skipped, even without specialized kernels. The amount of computation scales linearly (for MLP layers) or quadratically (for attention layers) with the number of tokens, and thus masking tokens can lead to a significant reduction in computation and time for the teacher forward.
- **Guides the teacher to teach better.** With the proposed masking strategy, the student accuracy *does not degrade* even when we mask 25–50% of all tokens. Interestingly, we find that the student performance sometimes even slightly increases when we remove only a small fraction of tokens; we give a more detailed description in Section 3.1.

We note that, unlike prior masking strategies for self-supervised learning, we do not mask the student. We find this critical for a better performance in supervised knowledge distillation; see Section 4.2.

## 3.1 MASKING STRATEGY

The primary aim of our masking strategy is enabling the teacher to provide a supervision that is more relevant and informative to the student's decision-making process. In other words, we want to prevent

the teacher from providing a misleading supervision by making a prediction on an image where all the core features—that the student uses for the prediction—have been masked.

To achieve this goal, we propose a student-informed patch saliency (SIPS) metric for masking the teacher input based on the student attention. SIPS utilize the attention scores from the last multi-head attention layer of the student, similarly to Caron et al. (2021); Chefer et al. (2022). We sum the attention scores from the class query to patch key vectors over all heads to get the SIPS score. We then mask a fixed fraction of image patches with the smallest scores.

More formally, suppose that the input image is divided into $N$ non-overlapping patches for both student and the teacher ViT.[1] Total $N + 1$ tokens, including the class token, is given as an input to the last multi-head attention layer of the student model. For the $h$-th attention head (among $H$ heads), the attention score from the class token to the image patch tokens are given as

$$\mathbf{a}^{(h)} = \text{Softmax}\left(\left(q_{\texttt{cls}}^\top k_1,\ q_{\texttt{cls}}^\top k_2,\ \cdots,\ q_{\texttt{cls}}^\top k_N\right)/\sqrt{d}\right), \qquad h \in \{1, 2, \ldots, H\} \qquad (2)$$

where $q_{\texttt{cls}}$ is the query of the class token, $k_i$ is the key of the $i$-th image patch token, and $d$ is the length of query and key vectors. Given this head-wise attention scores, we compute the SIPS metric by summing $\bar{\mathbf{a}} = \sum_{h=1}^{H} \mathbf{a}^{(h)}$. We use SIPS to select and mask $n$ patches with the smallest scores.

**Role of student guidance.** Through experiments, we find that using the student guidance is essential for performance (Section 4.2). Using SIPS, we often even observe an increased performance. We hypothesize that this is because masking often makes teacher make predictions more focused on core attributes instead of the context (e.g., background) that student cannot capture (right of Fig. 2). In other words, student-guided masking can help the teacher provide a better-tailored supervision.

**Computational efficiency.** The proposed SIPS introduces almost no computational overhead, as we can re-use the attention scores computed during the student forward. The only added computation is summing the attention scores, which takes $N \cdot (H - 1)$ FLOPs. The number of patches for ViTs is at most 576 (typically 196 or 256) and the number of heads is at most 48 (typically 16), and thus SIPS requires less than 28k FLOPs to be computed.

**Parallelization via micro-batching.** SIPS-based masking requires the student forward to precede the teacher forward. At first glance, this looks like an impediment toward parallel training, as it may introduce unnecessary idle time in teacher device. However, the problem can be easily resolved by adopting micro-batching (Huang et al., 2019), readily available in most deep learning frameworks such as PyTorch or TensorFlow; see Appendix E for details.

## 4 EXPERIMENTS

The experimental section is organized as follows:

- **Section 4.1** reports the performance of MaskedKD when applied for supervised ViT distillation on ImageNet-1k, over various choices of teacher and student models and base distillation algorithms.
- **Section 4.2** analyzes and validates design choices of MaskedKD and the SIPS metric, showing that (1) masking the teacher only, (2) using student guide to mask teacher, and (3) masking input instead of dropping tokens in the middle, are all essential components of MaskedKD.
- **Section 4.3** discusses the performance of MaskedKD over the boundary of supervised ViT distillation; in particular, we consider distilling the audio spectrogram transformer (Gong et al., 2021) and the distillation for the self-supervised training (Caron et al., 2021).

We also provide several additional experimental results in the appendix, including distillation from teachers trained with higher-resolution images (Appendix B), distilling only the linear classifiers (Appendix C), and measuring the effect of data augmentation in distillation (Appendix D).

### 4.1 MAIN EXPERIMENT

As our main setup, we consider the task of supervised ViT distillation with ImageNet-1k dataset (Russakovsky et al., 2015). We explore various choices of models and base distillation algorithms:

---

[1]When patch sizes differ, we interpolate the SIPS metrics bilinearly to compute patch saliency for the teacher.

Table 2: **MaskedKD on supervised ViT distillation.** MaskedKD dramatically reduces the supervision cost without degrading the student accuracy. "$\text{MaskedKD}_{\kappa\%}$" means that we keep only $\kappa\%$ of tokens from the teacher input. "Acc." denotes the ImageNet top-1 accuracy of the student model. "img/s" and "PFLOPs" denotes the throughput and the total supervision cost of the teacher throughout the training. ⚘ denotes that the model has an additional distillation token as an input; for this model, we do not report the performance of a model trained without distillation.

| Student | Teacher | Method | Acc. | img/s | PFLOPs |
|---|---|---|---|---|---|
| DeiT-Ti | - | No distillation | 72.0 | - | - |
| | DeiT-S | Logit | 75.0 | 1790 | 1770 |
| | | $+\text{MaskedKD}_{50\%}$ | 75.2 | 3702(x2.1) | 866(-51%) |
| | | $+\text{MaskedKD}_{40\%}$ | 74.9 | 4642(x2.6) | 707(-60%) |
| | | Manifold | 75.0 | - | - |
| | | $+\text{MaskedKD}_{75\%}$ | 75.2 | 2514(x1.4) | 1282(-28%) |
| | | Attention | 75.3 | - | - |
| | | $+\text{MaskedKD}_{50\%}$ | 75.3 | 3702(x2.1) | 866(-51%) |
| | DeiT3-S | Logit | 75.1 | - | - |
| | | $+\text{MaskedKD}_{50\%}$ | 75.2 | 3612(x2.0) | 866(-51%) |
| | | $+\text{MaskedKD}_{25\%}$ | 74.8 | 6894(x3.9) | 439(-75%) |
| | DeiT3-B | Logit | 74.4 | 750 | 6757 |
| | | $+\text{MaskedKD}_{50\%}$ | 74.7 | 1536(x2.0) | 3349(-50%) |
| | | $+\text{MaskedKD}_{25\%}$ | 74.7 | 2882(x3.8) | 1696(-75%) |
| | CaiT-S24 | Logit | 75.1 | 528 | 3591 |
| | | $+\text{MaskedKD}_{75\%}$ | 75.2 | 807(x1.5) | 2583(-28%) |
| | | $+\text{MaskedKD}_{50\%}$ | 75.2 | 1018(x1.9) | 1728(-52%) |
| | | Manifold | 75.7 | - | - |
| | | $+\text{MaskedKD}_{75\%}$ | 75.9 | 807(x1.5) | 2583(-28%) |
| | CLIP-B/16 | Logit | 73.9 | - | - |
| | | $+\text{MaskedKD}_{75\%}$ | 75.2 | 1017(x1.4) | 4932(-27%) |
| | | $+\text{MaskedKD}_{50\%}$ | 75.2 | 1536(x2.0) | 3349(-50%) |

| Student | Teacher | Method | Acc. | img/s | PFLOPs |
|---|---|---|---|---|---|
| DeiT-S | - | No distillation | 79.9 | - | - |
| | DeiT-B | Logit | 80.8 | 750 | 6757 |
| | | $+\text{MaskedKD}_{75\%}$ | 80.9 | 1038(x1.4) | 4932(-27%) |
| | | $+\text{MaskedKD}_{50\%}$ | 80.8 | 1536(x2.0) | 3349(-50%) |
| | DeiT3-B | Logit | 81.3 | - | - |
| | | $+\text{MaskedKD}_{75\%}$ | 81.4 | 1038(x1.4) | 4932(-27%) |
| | | $+\text{MaskedKD}_{50\%}$ | 81.3 | 1536(x2.0) | 3349(-50%) |
| DeiT-B | - | No distillation | 81.8 | - | |
| | DeiT3-L | Logit | 83.5 | 248 | 23677 |
| | | $+\text{MaskedKD}_{75\%}$ | 83.5 | 337(x1.4) | 17301(-27%) |
| | | $+\text{MaskedKD}_{50\%}$ | 83.6 | 512(x2.1) | 11744(-50%) |
| DeiT-S⚘ | MAE-ViT-B | G2SD | 81.5 | 750 | 4505 |
| | | $+\text{MaskedKD}_{75\%}$ | 81.6 | 1038(x1.4) | 3287(-27%) |
| MAE-ViT-L† | - | No distillation | 85.9 | - | - |
| | DeiT3-H-1k | Logit | 86.2 | 133 | 10723 |
| | | $+\text{MaskedKD}_{75\%}$ | 86.2 | 179(x1.3) | 7991(-25%) |
| | | $+\text{MaskedKD}_{50\%}$ | 86.1 | 271(x2.0) | 5301(-51%) |
| | DeiT3-H-21k | Logit | 86.5 | - | - |
| | | $+\text{MaskedKD}_{75\%}$ | 86.5 | 179(x1.3) | 7991(-25%) |
| | | $+\text{MaskedKD}_{50\%}$ | 86.3 | 271(x2.0) | 5301(-51%) |
| MAE-ViT-H† | - | No distillation | 86.9 | - | |
| | DeiT3-H-21k | Logit | 87.2 | - | - |
| | | $+\text{MaskedKD}_{75\%}$ | 87.2 | 179(x1.3) | 7991(-25%) |
| | | $+\text{MaskedKD}_{50\%}$ | 87.1 | 512(x2.0) | 5301(-51%) |

**Students.** We use ViTs of various sizes, trained from scratch using the training recipes of DeiT (Touvron et al., 2021a); DeiTs tend to perform better than vanilla ViTs, without requiring extra training dataset. For large-scale student models (ViT-L or ViT-H), we start from a self-supervised checkpoint trained with MAE, and fine-tune for 50 epochs (He et al., 2022).

**Teachers.** As a basic setup, we distill the knowledge from pre-trained ViT teachers that have larger sizes than the student. Additionally, we also consider various other scenarios: (1) CaiT (Touvron et al., 2021b): A model with slightly different architecture than ViT, (2) CLIP (Radford et al., 2021): A contrastively trained visual-language model that has been trained on a proprietary dataset not available to the student. (3) MAE (He et al., 2022): We distill from a fine-tuned MAE teacher, for comparison with distillation algorithms that require self-supervised teachers.

**Base distillation algorithms.** We use vanilla logit distillation as our default base distillation algorithm (Hinton et al., 2015), as it is most widely used and generally applicable algorithm (even for proprietary teachers). Additionally, we consider the following base distillation methods:

- *Manifold* (Hao et al., 2022): The manifold distillation regularizes the student features to share the same patch-level manifold structure with the teacher features.

- *Attention* (adapted from Wang et al. (2022)): Attention distillation transfers the teacher's class-patch attention scores to the student. Unlike Wang et al. (2022), we apply this technique to the supervised distillation. In the supervised setup, we find that distilling attention scores of all layers perform better than distilling only the last layer; we distill all layers in this paper.

- *G2SD* (Huang et al., 2023): G2SD is a two-stage algorithm which first distills from a self-supervised teacher (trained via MAE) and then from a supervised teacher obtained by fine-tuning the self-supervised teacher on the target task. We apply MaskedKD during the fine-tuning stage only.

**Patch size.** By default, each $224 \times 224$ image is divided into total 196 patches of size $16 \times 16$ pixels. The DeiT3-H and MAE-ViT-H models use 256 patches of size $14 \times 14$. Whenever there is a mismatch between the number of patches between the teacher and the student, we bilinearly interpolate the SIPS score on student patches and to compute SIPS for teacher patches.

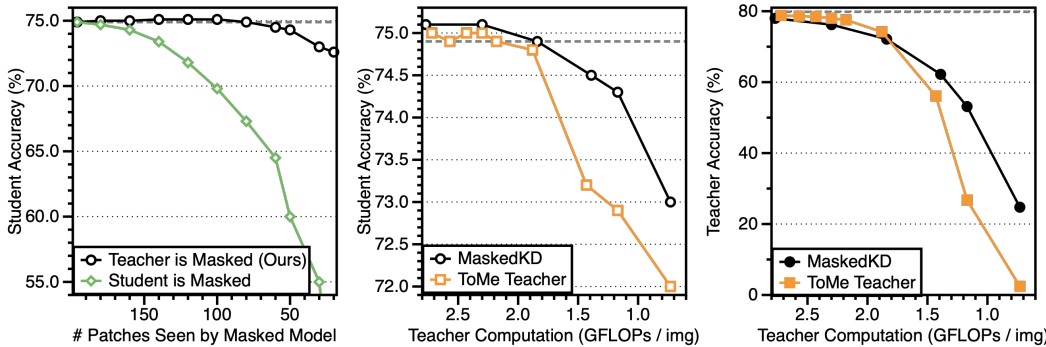

Figure 3: (⇐) **Masking teacher vs. student.** Masking the student during distillation substantially degrades the student accuracy, but masking the teacher does not. (⇑, ⇒) **MaskedKD vs. ToMe.** Applying ToMe to the teacher also reduces the supervision cost, but MaskedKD achieves a better accuracy-computation tradeoff than ToMe (⇑). This happens both in high-compute and low-compute regime, regardless of which teacher predicts better (⇒).

**Seeds.** We averaged over 3 trials, except for the CLIP experiments and large/huge student models.

**Other details.** Other experimental details are given in Appendix A.

**Result and discussion.** In Table 2, we provide the performances of MaskedKD when applied to various supervised distillation scenarios.[2] From the table, we observe that we can safely remove 25–50% of the patches from the teacher input, without sacrificing the student accuracy; in some cases, we can even remove 60–75% of the patches. We also observe that, in most cases, masking a small fraction of patches is beneficial for the performance of the trained student (although with a very small boost). Such performance gain from masking is most pronounced when the size gap between the teacher and the student is large (DeiT3-B → DeiT-Ti). From this observation, we hypothesize that the masking may have an effect similar to making a (low-capacity) *teaching assistant* model (Mirzadeh et al., 2020), which can teach the student better than an overly large teacher.

## 4.2 A CLOSER LOOK AT THE MASKING STRATEGY

Here, we validate several design choices of MaskedKD. All experiments on have taken place on the task of distilling the knowledge of DeiT-S to DeiT-Ti, using the ImageNet-1k dataset.

**Mask teacher, not student.** In the left of Fig. 3, we compare the student accuracy of MaskedKD with its variant where the student is masked instead of the teacher. We observe that masking the student during distillation immediately degrades the student performance even at a very low masking rate. In contrast, masking the teacher (as in MaskedKD) does not degrade the student accuracy until one masks over 50% of all patches. This illustrates the crucial difference on the role of masking in supervised distillation and self-supervised learning, e.g., Assran et al. (2022); Chen et al. (2022).

**Mask at input, not in the middle.** In the middle and left of Fig. 3, we compare the accuracy of students supervised by masked teacher with the students supervised by teachers accelerated with token merging (Bolya et al., 2023). We observe that MaskedKD outperforms ToMe-based distillation, even without having to modify the structure of the teacher model (thus applicable to distillation from proprietary teachers). Interestingly, in the high-compute regime, the student accuracy of MaskedKD is higher in spite of the lower accuracy of the teacher model itself. This observation suggests that MaskedKD teacher retains the supervision quality better than ToMe teacher, by having a view aligned with the student. In the low-compute regime, the teacher also predicts better. This is because the gradual merging strategy of ToMe forces a more extreme merging at the last layer than input masking to achieve a similar level of teacher computation reduction.

---

[2]For G2SD, the student accuracy we obtained for the baseline (81.5%) is slightly lower than what is reported in the original paper. We could not reproduce the reported accuracy although we used the official code.

Table 3: **Validating SIPS.** We ablate various components of SIPS: Extracting [(a)]class-patch tokens from the [(b)]last layer of the [(c)]student model, and removing all patches except for [(d)]top-k SIPS score. We use DeiT-S as a teacher and DeiT-Ti as a student, and mask away 50% of all input image patches. The default MaskedKD setting is marked in purple .

(a) **Class or patch token?**

| method | top-1 | top-5 |
|---|---|---|
| [cls]-patch | **75.1** | **92.1** |
| patch-patch | 74.6 | 91.9 |

(b) **Which layer?**

| method | top-1 |
|---|---|
| first (1) | 72.3 |
| middle (6) | 75.0 |
| last (12) | **75.1** |

(c) **Student vs. DINO**

| select | top-1 | flops |
|---|---|---|
| None | 74.9 | 4.6 |
| DINO | 75.0 | 6.9 |
| student | **75.1** | **2.3** |

(d) **Random & Bottom-$k$**

| function | top-1 | top-5 |
|---|---|---|
| top-k | **75.1** | **92.1** |
| random | 74.6 | 91.3 |
| bottom-k | 71.7 | 80.0 |

**SIPS to guide teacher what to teach.** In Table 3, we ablate various components of the SIPS score. From the table, we observe that:

(a) Using the class-patch attention leads to a better masking than using (an average of) the patch-patch attention score. Using class-patch attention also introduces less computational overhead.

(b) Using the attention score from the last layer works better than using the attention score from preceding layers. This observation is well-aligned with the observation by Caron et al. (2021) that the attention becomes more focused and semantic in the last transformer block.

(c) Using the SIPS score from the student works better than using the attention of external models, such as DINO (Caron et al., 2021). Using the student attention is also computationally efficient, as it does not require additional computation.

(d) The SIPS score is well-correlated with the accuracy of the student. Making the teacher predict on bottom-$k$ patches introduces a large degradation in the student performance.

### 4.3 EXTENDING THE BOUNDARIES OF MASKEDKD

Here, we test the performance of the MaskedKD for the tasks other than the supervised ViT distillation. In particular, we apply MaskedKD to (1) the distillation of audio spectrogram transformers, and (2) the self-supervised learning algorithms that utilize some form of self-distillation.

**Distilling audio transformers.** Here, we check whether the MaskedKD can also be applied for an efficient distillation of the transformer that processes data from other domains. To this end, we consider distilling an audio spectrogram transformer (AST) (Gong et al., 2021) on the ESC-50 audio classification dataset (Piczak, 2015). AST has an identical architecture to DeiT, but uses a modified training recipe and hyperparameters tailored for processing the spectrograms of the speech data. We give a more detailed description of the experimental setups in Appendix A.

Table 4 compares the performance of the vanilla logit distillation algorithm against the distillation with MaskedKD. We observe that the MaskedKD can indeed reduce the number of patches used by 17% without sacrificing the performance (masking 100 patches out of 600). The supervision cost reduction, however, is not as much as in the vision domain. The performance drops by 0.6% if we use only 50% of the patches, unlike most vision transformers.

**Self-distillation for self-supervised learning.** Here, we apply MaskedKD to reduce the computations of a distillation-based self-supervised learning algorithm, DINO (Caron et al., 2021). A line of self-supervised learning literature aims to train useful image representations by distilling the knowledge from a teacher model—generated as a moving average of the student—that sees a differently augmented version from the student (Grill et al., 2020). Since its advent, such *self-training* algorithms for self-supervised learning has been one of the major use cases of the knowledge distillation. In the context of ViT, DINO (Caron et al., 2021) is one of the most prominent algorithms in the direction. DINO employs a teacher that sees the full image, a student that sees the full image as well, and additional students having smaller field of views (FOVs); then, DINO performs the self-distillation to regularize the teacher and students to give similar outputs.

We apply MaskedKD to DINO as follows: We use only the attention scores of a student that sees the full image to compute the patch saliency, and mask the teacher input. As the student models with smaller FOVs have a much smaller computational cost for inference, masking the teacher can save a substantial portion of the whole training cost.

Table 4: **Audio classification.** We apply MaskedKD on distilling the audio spectrogram transformer (Gong et al., 2021) for the audio classification on ESC-50 (Piczak, 2015).

| Student | Teacher | Method | Acc. (%) |
|---------|---------|--------|----------|
| AST-S | - | No distillation | 85.1 |
| | AST-B | Logit | 86.3 |
| | | +MaskedKD $_{83\%}$ | 86.4 |
| | | +MaskedKD $_{50\%}$ | 85.7 |

Table 5: **Masking DINO.** We apply MaskedKD on the self-supervised training procedure of DINO (Caron et al., 2021). "Linear" denotes the linear probing accuracy and "PFLOPs" denotes the total teacher FLOPs.

| Model | Method | Linear (%) | PFLOPs |
|-------|--------|------------|--------|
| DeiT-S | DINO | 73.7 | 589 |
| | +MaskedKD $_{87\%}$ | 73.9 | 507 |
| | +MaskedKD $_{77\%}$ | 74.0 | 446 |
| | +MaskedKD $_{66\%}$ | 74.2 | 384 |
| | +MaskedKD $_{61\%}$ | 73.6 | 354 |

Table 5 compares the performance of the vanilla DINO against MaskedKD, where we use ImageNet-1k dataset for both the pre-training phase (100 epochs) and fine-tuning phase. We observe that we can mask away more than 30% of the patches from the teacher input without a degradation in the quality of the learned representation; the quality of representation is measured by the accuracy achievable by linear probing, i.e., only the linear classifier is fine-tuned for the task. We can save the teacher computation accordingly, cutting down the total teacher FLOPs from 589PFLOPs to 384PFLOPs when we use 66% of the patches.

## 5 Discussion

In this work, we highlight the computational overhead of the knowledge distillation procedure, and propose an algorithm that can reduce the computational burden of ViT distillation. In particular, we have introduced MaskedKD, a method that masks out a fraction of input patches given to the teacher ViT model; this cuts down the teacher forward cost dramatically, by allowing us to skip the computations associated with the masked patches. To select the patches to discard, we devise a patch saliency measure which is designed to prevent the teacher from giving supervisions on the image where all the core features (that the student uses for prediction) have been masked away. The proposed patch saliency is computed by re-using the attention score from the student ViT's last multi-head attention module, and thus introduces only a minimal computational overhead. In our experiments, we have applied the proposed MaskedKD to many ViT distillation scenarios, showing that we can reduce the number of patches the teacher sees by 25–50% without any student performance drop.

**Limitations and future directions.** A major limitation of the present work is that the proposed MaskedKD is specialized to the transformer-based models. The main mechanism behind the computation and latency savings is the token reduction, which can only be applied to transformer-like models which process the input as a sequence of tokens. Our masking strategy also exploits the information that have been generated by the attention mechanism of the transformer architecture. Generalizing the current framework to cover a broader range of networks including, e.g., convolutional networks or recurrent neural networks, is an important future research direction.

Another big limitation of the current work is that we put our focus on distillation of models trained by *supervised learning*. Modern large-scale models are typically trained via self-supervised learning, and the knowledge distillation often takes place at this stage. While the core underlying idea of the MaskedKD can be applied to the models trained with self-supervised learning, and MaskedKD can be used to make some distillation-based self-supervised learning algorithms more efficient (as demonstrated in Section 4.3), it is in general unclear how one can combine it with other self-supervision strategies that utilize masking, e.g., masked autoencoders (He et al., 2022).

Lastly, we note that MaskedKD is in need of a mechanism to efficiently find the optimal number of tokens to use for the teacher input. The goal of MaskedKD is to reduce the training computation; if one can reduce the training compute only after multiple trial runs, the advantage of the method becomes unclear. While we empirically find that "50%" may be a working rule-of-thumb for the tasks in the vision domain, the empirical results on the audio data suggest that the optimal fraction may differ from a domain to domain.

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

## A  DETAILS ON THE EXPERIMENTAL SETUP

**Training recipe and data augmentations.** For training student ViTs, we follow the settings of DeiT (Touvron et al., 2021a), except for the tiny model; we find that Tiny ViT tends to achieve better accuracy with less data augmentations, similar to Steiner et al. (2022). In particular, we only use random resized crops and horizontal flips for tiny model, without RandAugment, *mixup* and *cutmix*.

**Hardware.** Throughputs reported in this paper are measured on a single NVIDIA RTX A6000 graphic card using FP32 weights/activations and the batch size 128.

**Distillation hyperparameters.** For the balancing hyperparameter $\lambda$ and the temperature scale $\tau$ (for logits), we use $(1.0, 1)$ unless other noted otherwise; we have tuned the hyperparameters over the search space $\{0.1, 1.0, 9.0\} \times \{1, 2, 3, 4\}$, and observe that $(1.0, 1)$ works well throughout all setups. Similar observations about the hyperparameters have been made in Zhang et al. (2022); Wu et al. (2022a); Hao et al. (2022).

**Manifold distillation.** While the original paper only uses CaiT as the teacher model, we also experiment with DeiT teachers; we use the same hyperparameters for training with DeiT teachers.

**Attention distillation.** As briefly discussed in the main text, we adapt the attention distillation to the supervised learning setup by distilling the attention from all layers (with the same scaling factors), rather than distilling only the last layer.

**G2SD.** We apply MaskedKD only during the supervised distillation stage of the G2SD.

**Audio experiments.** We use the official code of AST[3] and follow the same training procedure; the model's performance is evaluated by averaging 5 seed validation results.

**DINO.** We follow the smaller-scale experimental setup for DINO, available at the official code repository[4], where we train for 100 epochs. We also halved the number of GPUs (from 8 to 4) and the per-GPU batch size (from 256 to 128), due to the limitations in the computing resource available.

---

[3] https://github.com/YuanGongND/ast
[4] The "vanilla DINO training" in https://github.com/facebookresearch/dino/

Table 6: **MaskedKD with teachers that take higher-resolution images.** We validate the effectiveness of MaskedKD with the teacher model that takes in higher-resolution images as inputs. Teacher models are trained with $384 \times 384$ resolution images.

| Student | Teacher | Method | Acc. | PFLOPs |
|---------|---------|--------|------|--------|
| DeiT-Ti | DeiT3-S @ 384 | Logit | 74.9 | 5694 |
| | | +MaskedKD$_{75\%}$ | 75.3 | 4270 (-28%) |
| | | +MaskedKD$_{50\%}$ | 74.9 | 2725 (-54%) |
| | | +MaskedKD$_{34\%}$ | 74.6 | 1814 (-70%) |
| | DeiT3-S @ 224 | Logit | 74.9 | 1771 |
| | | +MaskedKD$_{50\%}$ | 75.1 | 866 (-51%) |
| | | +MaskedKD$_{25\%}$ | 74.8 | 439 (-75%) |

## B  TEACHERS TRAINED WITH HIGHER-RESOLUTION IMAGES

One of the key factors that govern the inference compute of a model is the *input resolution.* While the models that are trained on higher-resolution images tend to work better (Touvron et al., 2022), the models tend to require much higher computational cost. We ask whether the proposed MaskedKD can be used to distill the knowledge from the teachers that are trained on higher-resolution images, without inducing an excessive overhead in the training cost.

To this end, we consider the following MaskedKD pipeline for distilling from the ViT teachers that takes a higher-dimensional input.[5] More specifically, we consider distilling ViTs trained with $384 \times 384$ images, which are processed by dividing into total 576 patches of size $16 \times 16$.

(1) Given a low-resolution image, forward the image through the student model to get the predictions and the mean attention scores for the patches; here, if we use $224 \times 224$ input image, then we get 196 scores.

(2) Interpolate the mean attention scores to generate the scores for larger number of patches (e.g., 576 patches). We use bilinear interpolation for our experiment.

(3) Interpolate the input image to generate a high-resolution image, and mask the image with the interpolated mean attention scores computed in the previous step.

(4) Forward the masked image through the teacher model and use the prediction to regularize the student training.

We give the experimental results in Table 6. Here, we observe that one can successfully distill the knowledge from a teacher trained with higher-resolution images. Also, we observe that MaskedKD successfully reduces the computational burden for distillation, without any performance drop. We note, however, that we did not count the FLOPs that are required for the interpolation procedures in step (2,3). Also, the forward FLOPs for the teacher trained with higher resolution images are too big, so that the performance of the MaskedKD-distilled model does not match the performance of the model distilled using vanilla KD + low-resolution teacher that has a similar training FLOPs.

---

[5]We note that, up to our knowledge, this is the first attempt distilling a high-resolution teacher to a low-resolution student.

Table 7: **MaskedKD for distilling only the linear classifiers.** We apply MaskedKD to a scenario where we only fine-tune the linear classifier of the student, whose (frozen) feature map has been pre-trained with self-supervision.

| Student | Teacher | Method | Acc. | PFLOPs |
|---|---|---|---|---|
| DINO-ViT-S | - | No distillation | 76.9 | - |
| | DeiT-B | Logit | 77.5 | 2252 |
| | | +MaskedKD $_{75\%}$ | 77.5 | 1644(-27%) |
| | | +MaskedKD $_{50\%}$ | 77.5 | 1116(-49%) |
| | DeiT3-L | Logit | 77.5 | 7892 |
| | | +MaskedKD $_{75\%}$ | 77.6 | 5767(-27%) |
| | | +MaskedKD $_{50\%}$ | 77.5 | 3914(-50%) |
| DINO-ViT-B | - | No distillation | 77.9 | - |
| | DeiT-B | Logit | 78.1 | 2252 |
| | | +MaskedKD $_{75\%}$ | 78.1 | 1644(-27%) |
| | | +MaskedKD $_{50\%}$ | 78.1 | 1116(-49%) |
| | DeiT3-L | Logit | 78.2 | 7892 |
| | | +MaskedKD $_{75\%}$ | 78.3 | 5765(-27%) |
| | | +MaskedKD $_{50\%}$ | 78.2 | 3914(-50%) |

## C  MASKEDKD FOR DISTILLING LINEAR CLASSIFIERS OF SELF-SUPERVISED MODELS

We also consider the scenario where we distill the knowledge of a supervisedly trained teacher to a student whose feature map has been pre-trained with a self-supervised learning scheme and frozen; during the distillation, we only fine-tune the linear classifier of the student. In such scenario, reducing the teacher computation gains even greater importance, as the computational cost for the student backward is greatly diminished. Previous studies on knowledge distillation primarily focuses on cases where the entire model is fine-tuned. However, several recent studies show that fine-tuning the entire model may be suboptimal in some cases (Lee et al., 2023; Park et al., 2023).

Table 7 provides the experimental results. We observe that KD still provides performance boost under this setup, and the efficiency gain of MaskedKD takes place again. One interesting observation is that using a large teacher (DeiT3-L) for a relatively much smaller student (ViT-S) does not degrade the performance in this case, unlike in the typical case where we fine-tune all layers of the student model.

Table 8: **Flexibility in data augmentation.** Our method overcomes FastKD's limitation, i.e., being restricted to simple data augmentations, resulting in an improved student model performance. "Simple" refers to applying basic augmentations: random resized crop and horizontal flip. "Hard" means additionally performing RandAugment, *mixup* and *cutmix*.

| Student | Augmentation | Teacher | Method | Acc. | PFLOPs |
|---------|--------------|---------|--------|------|--------|
| DeiT-S | Simple | - | No distillation | 71.3 | - |
| | | DeiT-B | Logit | 79.7 | 6757 |
| | | | +MaskedKD $_{50\%}$ | 80.2 | 3349(-50%) |
| | Hard | - | No distillation | 79.9 | - |
| | | DeiT-B | Logit | 80.8 | 6757 |
| | | | +MaskedKD $_{50\%}$ | 80.9 | 3349(-50%) |

## D  MASKEDKD AND DATA AUGMENTATIONS

One of the key advantages of the MaskedKD comparing with the FastKD (Shen and Xing, 2022) is that MaskedKD can be applied to distillation scenarios where we use heavy data augmentation schemes. As FastKD requires pre-computing and storing the teacher predictions for all augmented samples, computational benefits of the FastKD may be greatly undermined by considering heavier and more diverse augmentations. In this section, we perform a basic sanity check that (1) such heavy data augmentations are indeed useful in KD scenarios,[6] and (2) MaskedKD works well with heavy augmentations.

Table 8 gives the experimental results. We find that, even for relatively small-scale student models such as DeiT-S, the data augmentation greatly boost the model performance. For undistilled students, the gain can be as large as $8.6\%p$. For the models trained with basic logit distillation, the gain is $1.1\%p$. We also observe that MaskedKD preserves the student model accuracy with both light and heavy data augmentations.

---

[6]Under non-KD contexts, Steiner et al. (2022) makes a similar observation.

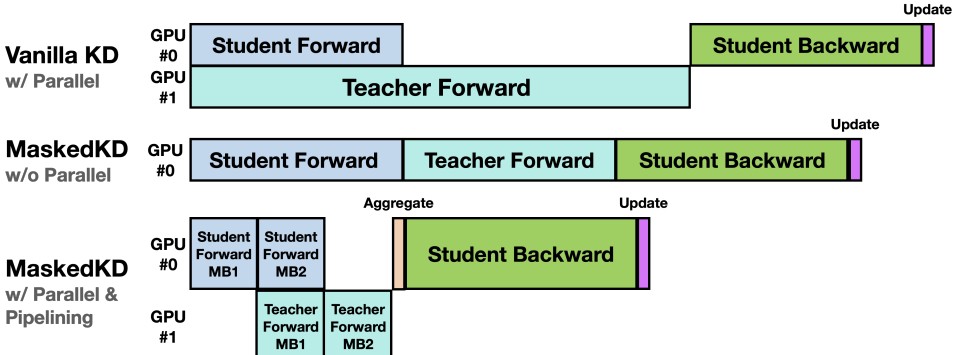

Figure 4: **Right : Training time breakdown of MaskedKD.** (Top): In vanilla KD, the student and teacher forwards can be processed in parallel. (Middle): In MaskedKD, the teacher forward may wait until the student forward ends, and yet, the teacher forward can be reduced a lot. (Bottom): Pipeline parallelism allows MaskedKD to efficiently utilize multiple GPUs.

## E  PIPELINING MASKEDKD

To apply the MaskedKD, we need to compute the patch saliency before the teacher inference stage. As the saliency score is computed at the last layer of the student, the teacher forward cannot begin until the student forward is completed. Thus, a naïve parallelization strategy of running the teacher and the student model on two seperate GPUs may be somewhat less effective than in the vanilla knowledge distillation (Fig. 4, top).

However, this *does not* imply that (1) there is no speedup, or (2) there is no effective parallelization strategy. First, we note that the teacher forward is usually the key bottleneck in knowledge distillation, often taking much longer than the student forward. MaskedKD dramatically reduces the time for teacher forward, so that the MaskedKD running the student and teacher forward in series can be faster than the teacher forward of the vanilla KD in some cases (Fig. 4, middle). Second, when using multiple GPUs, we can use pipelining to fill up the bubbles, as in GPipe (Huang et al., 2019). More specifically, one can divide each training data batch into smaller mini-batches and make forward inferences on them sequentially. This division allows the teacher GPU to access the data before the student forward completes on all mini-batches (Fig. 4, bottom).

# F  LARGE TEACHER WITH MORE MASKING VS. SMALL TEACHER WITH LESS MASKING

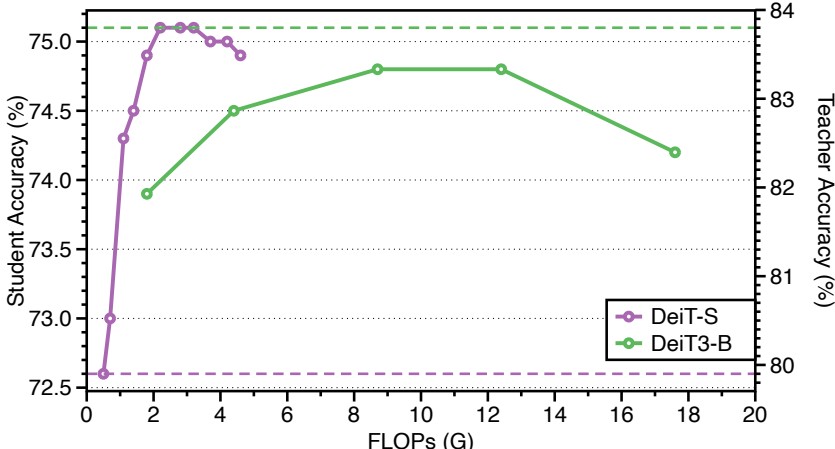

Figure 5: **Differences of the MaskedKD results by model size.** We compare two teachers: DeiT-S and DeiT3-B. The colored dashed lines denote the accuracy of the teacher model. The performance of the teacher model is depicted by the dashed line, indicating its level of performance (Right). The line with dots represents the performance of the distilled student model performance (Left).

Masking the teacher input gives a new way to trade the student performance for the training efficiency, in addition to changing the model size. In this section, we compare the performance-efficiency tradeoff curve of two different-sized teacher models. In particular, we examine whether a larger teacher that uses less patch can give a more cost-efficient guidance than a smaller teacher that uses more patches. For this purpose, we compare the performance of the DeiT-Tiny students trained with DeiT-S and DeiT3-B teachers that use varying fraction of patches (Fig. 5). We observe that masking does not give a dramatic change to the answer to the question "which teacher is most cost-efficient?" DeiT-S teacher dominates DeiT3-B teacher at all per-iteration FLOPs level.

# G IMPLEMENTATION

The following is the pseudo code of our "MaskedKD Engine" in PyTorch (Paszke et al., 2019):

```python
def maskedkd_engine(image, labels, student, teacher, num_keep):

    '''num_keep: the int number of patches to keep.'''

    output, attn = student(image)

    len_keep = torch.topk(attn.mean(dim=1)[:,0,1:],num_keep).indices

    def teacher_inference(image, teacher, len_keep):
        x = teacher.patch_embed(image)
        B, _, D = x.shape   # batch, length, dim

        cls_tokens = teacher.cls_token.expand(B, -1, -1)
        x = torch.cat((cls_tokens, x), dim=1)
        x = x + teacher.pos_embed

        cls_save = x[:, 0, :].unsqueeze(dim=1)
        x = x[:, 1:, :]
        index = len_keep.unsqueeze(-1).repeat(1, 1, D)
        x = torch.gather(x, dim=1, index= index)
        x = torch.cat((cls_save, x), dim=1)

        for blk in teacher.blocks:
            x = blk(x)

        x = teacher.norm(x)
        output = teacher.head(x)
        return output

    t_output = teacher_inference(image, teacher, len_keep)

    loss = CE(output, labels) + KL_DIV(output, t_output)

    return loss
```

This returns the loss. By simply adding the patch selection stage, we can greatly improve efficiency and achieve substantial gains.

# H FLOPS OF A TRANSFORMER BLOCK

Table 9: **Analyzing Computational Cost.** We examine how the computational cost varies when applying MaskedKD. The arrow symbol represents the difference in FLOPs when utilizing MaskedKD.

| Layer | Complexity | Computation (GFLOPs) | | |
|---|---|---|---|---|
| | | DeiT-S @ 384 | DeiT-B | DeiT-S |
| Softmax-Attention | $\mathcal{O}(LN^2M)$ | $3.1 \to 0.8$ | $0.7 \to 0.2$ | $0.4 \to 0.1$ |
| Projections | $\mathcal{O}(LNM^2)$ | $4.1 \to 2.0$ | $5.6 \to 2.8$ | $1.4 \to 0.7$ |
| MLP | $\mathcal{O}(LNM^2)$ | $8.2 \to 4.1$ | $11.2 \to 5.6$ | $2.8 \to 1.4$ |
| Total | $\mathcal{O}(LNM(M+N))$ | $15.3 \to 6.9$ | $17.5 \to 8.6$ | $4.5 \to 2.2$ |

We analyze the FLOPs of Vision Transformer (ViT) in this section. ViT is encoder-only transformer Vaswani et al. (2017), which is mainly consisted of a multi-head self-attention layers and a multi-layer perceptron layers. There are also many details which takes very tiny portion in terms of calculation, such as the embedding layer, residual connection, bias, GeLU, or layer normalization and we will ignore it in this section. We denote $\phi(n, d)$ as a function of FLOPs with respect to the number of tokens $n$ and the embedding dimension $d$. For example, in case of DeiT-B, n is 197 and d is 768. For self-attention layer, the FLOPs mainly comes from two parts: (1) The projection of $Q,K,V$ matrices and the self-attention outputs $\phi_{\text{proj}}(n, d) = 4nd^2$, (2) The calculation of the softmax-attention $\phi_{\text{SA}}(n, d) = 2n^2d$.

The FLOPs of MLP layers comes from two fully-connected (FC) layers. Two FC layers have a difference of four times in dimension. Therefore, the FLOPs for MLP layer is $\phi_{\text{FC}}(n, d) = 8nd^2$.

By combining the self-attention layer and the MLP layer, we can get the total FLOPs of one ViT block.

$$\phi_{\text{BLK}}(n, d) = \phi_{\text{proj}}(n, d) + \phi_{\text{FC}}(n, d) + \phi_{\text{SA}}(n, d) = 4nd^2 + 2n^2d + 8nd^2 = 12nd^2. \quad (3)$$

Since there is 12 layers in case of DeiT-Base, the total FLOPs is

$$12 * \phi_{\text{BLK}}(n, d) = 12 * (12nd^2 + 2n^2d) = 144nd^2 + 24n^2d. \quad (4)$$

