# OpenReview forum: "MaskedKD: Efficient Distillation of Vision Transformers with Masked Images"
_ICLR.cc/2024/Conference — ICLR 2024 Conference Withdrawn Submission_

### Official Review · Reviewer_tQVJ · 2023-10-18

**Soundness:** 3 good
**Presentation:** 4 excellent
**Contribution:** 2 fair
**Rating:** 5
**Confidence:** 4

**Summary:**

The paper aims to reduce the training cost of knowledge distillation. The paper determines the informative patches based on the student network's attention and only send these informative patches to the teacher network. This reduces the computational load during training.

**Strengths:**

1. The writing in this paper is very clear, the method is simple and effective, and it is easy to follow.
2. The experiments in the paper demonstrate that the proposed method can reduce training cost without sacrificing the student's accuracy.

**Weaknesses:**

1. Generality. It seems that this approach can only be applied to ViT-like structures, which limits the widespread applicability of the method.
2. Novelty. I believe that the MAE[1] has already informed the commuity that the ViT architecture can reduce training costs by using only a portion of the patches. Therefore, I personally think that the novelty of this paper seems to be insufficient, as it merely verifies something that is already well-known.
3. Result. I guess that the slight improvement in the results could also be attributed to the use of a stronger augmentation technique, such as masking.

[1]. K. He, X. Chen, S. Xie, Y. Li, P. Dollár, and R. Girshick. Masked autoencoders are scalable vision learners. In Proceedings of the IEEE/CVF Conference on Computer Vision and Pattern Recognition, 2022

**Questions:**

See weaknesses.

---

### Official Review · Reviewer_GaAs · 2023-10-29

**Soundness:** 3 good
**Presentation:** 3 good
**Contribution:** 2 fair
**Rating:** 5
**Confidence:** 4

**Summary:**

This paper proposes a distillation method named MaskedKD, which cuts down the ViT supervision cost quite dramatically without degrading student accuracy. It selects important tokens according to the student's feature and then feeds the tokens to the teacher model, saving much training time.

**Strengths:**

1. MaskedKD just needs partial patch tokens for the teacher model's inference, which saves much training time and resources.
2. At the same time, the student's performance remains the same and even gets higher, which is interesting.
3. The results on various datasets and models prove its effectiveness. Besides, MaskedKD can be combined with other methods.

**Weaknesses:**

1. Some reference papers about masked strategy or distillation for ViTs are missing:

     [1] Masked Autoencoders Are Stronger Knowledge Distillers.

     [2] Masked autoencoders enable efficient knowledge distillers.

     [3] Vitkd: Practical guidelines for vit feature knowledge distillation.

     [4] Dearkd: data-efficient early knowledge distillation for vision transformers.
2. Although masking the input images as the teacher's input is interesting, a similar idea has been applied in DMAE. In this way, the main contribution seems to lack value.

     DMAE: Masked autoencoders enable efficient knowledge distillers.
3. This paper takes logit-based distillation as the final loss. I am curious about if MaksedKD can be applied for feature-based (e.g ViTKD) and logit-based methods (KD) together. Could the student achieve higher performance?
4. This paper takes the traditional KD method. How about the latest DIST, DKD, NKD? Does MaskedKD still work well?

    [1] DIST: Knowledge distillation from a stronger teacher.

    [2] DKD: Decoupled Knowledge Distillation.

    [3] NKD: From Knowledge Distillation to Self-Knowledge Distillation: A Unified Approach with Normalized Loss and Customized Soft Labels

**Questions:**

see above

---

### Official Review · Reviewer_WM1F · 2023-11-01

**Soundness:** 2 fair
**Presentation:** 2 fair
**Contribution:** 2 fair
**Rating:** 5
**Confidence:** 5

**Summary:**

To reduce the cost of supervision when distilling from large-scale models, this paper proposes using only a subset of the most salient patches for the teacher model. Only a fraction of the image patch tokens are fed to the teacher, bypassing the computations needed to process those patches. The masking mechanism is based on the attention score of the student. Results show that in some instances, up to 50% of the patches can be omitted from the teacher's input without affecting the student's accuracy.

**Strengths:**

- The idea of masking teacher input to reduce distillation cost is intriguing.

- In the supervised setting, this masking scheme can lead to faster distillation speeds while maintaining the student's performance.

- This work also explores MaskedKD outside the visual domain, such as in audio spectrogram transformers.

**Weaknesses:**

- The reason why student-informed patch saliency works is not clear. It would be good to explore and explain the underlying mechanism of this scheme.
- Is the student model utilized for patch selection at the onset of training? If it is, why would a student model be advantageous in the initial stages? If not, how many training epochs are required for the student model to select the appropriate patches?

- Several studies have leveraged the masking scheme for efficient learning. For instance, references [1,2] demonstrate that mask-prediction distillation can lead to quicker convergence and enhanced performance. In contrast, this study emphasizes masking teachers to reduce training expenses. I'm curious to know if the proposed method is more cost-effective than the mask-prediction schemes in terms of overall training costs. Additionally, given that these techniques allow for a significantly higher masking ratio for the student network, it's ambiguous whether the savings incurred for the teacher model in MaskedKD surpass the savings for the student model in these studies.

[1] Milan: Masked Image Pretraining on Language Assisted Representation
[2] Towards Sustainable Self-supervised Learning

- This work only shows the classification performance on ImageNet. I would expect more results on downstream tasks such as semantic segmentation on ADE20k dataset.

**Questions:**

Please follow the questions in the weakness section. My major questions are
-  the advantage over mask-prediction distillation methods,
-  the underlying mechanism  of student-informed patch saliency.

---

### Official Review · Reviewer_GJ8d · 2023-11-01

**Soundness:** 3 good
**Presentation:** 4 excellent
**Contribution:** 3 good
**Rating:** 6
**Confidence:** 4

**Summary:**

This paper introduces MaskedKD, a new distillation approach designed to accelerate Transformer-based distillation without incurring (obvious) performance drop. Specifically, MaskedKD proposes dropping patches of the input image to the teacher model, while keeping the input to the student model intact. This approach saves the compute by a non-trivial margin, and empirically does not lead to student performance degradation. The student attention score with respect to the classification token is used to generate mask. Experiments on the commonly used vision dataset ImageNet and a audio dataset ESC-50 demonstrate the effectiveness of the proposed MaskedKD.

**Strengths:**

1. Applying the idea of masking partial input images in KD intuitively makes sense to me. Images are inherently redundant natural signals, and it has already been shown in many Masked Image Modeling (MIM) works that only a fraction of the image is enough to reconstruct the whole image at a reasonable quality. In this sense, only a part of the image should also suffice for knowledge distillation.
2. The writing of organization of this paper is good. its logic follows a reasonable thread.
3. The ablation study is comprehensive and overall in good quality. Significant experimental details are given, and it should be relatively easy to reproduce this work.
4. The experiments on the audio dataset add extra points for this submission, showing the generalizability for Masked KD.

**Weaknesses:**

1. I find the improvement of distillation is limited when both teacher and student models are large. For example, in table 2, using MAE-ViT-H as student, and DeiT-3-H-21k as teacher, only leads to 0.2%-0.3% improvement. Is it because of the wrong choice of teacher-student pair, or not so advanced distillation method? The authors should investigate into this issue.
2. Following the last one, the gap of most of the teacher-student pairs are only one level (e.g. base2small, large2base). The only exception is base2tiny. The authors should try more teacher-student combinations, like large2small, or even huge2small.
3. The authors are encouraged to try both more classical feature-based method[1] and more recent distillation methods [2][3].


References

[1] Romero, Adriana, et al. "Fitnets: Hints for thin deep nets." arXiv preprint arXiv:1412.6550 (2014).

[2] Chen, Xianing, et al. "DearKD: data-efficient early knowledge distillation for vision transformers." Proceedings of the IEEE/CVF Conference on Computer Vision and Pattern Recognition. 2022.

[3] Ren, Sucheng, et al. "Co-advise: Cross inductive bias distillation." Proceedings of the IEEE/CVF Conference on computer vision and pattern recognition. 2022.

**Questions:**

1. In the right figure of figure 2, the authors provide some intuitive explanation as to why use the student attention in mask generation, rather then teacher attention, to avoid misleading supervision. However, there is no corresponding expeirments in ablation study. I wonder what if teacher attention is used in MaskedKD?

---

### Author Response · Authors · 2023-11-23
**We appreciate your reviews.**

Dear reviewers and AC,

We sincerely appreciate your precious time and effort on reviewing our manuscript. We have decided to withdraw the paper---we have made some new observations in preparation of our response, which we believe will lead to an improved version of the proposed method.

Best regards,
Authors.